# *Cannabis*: From a Plant That Modulates Feeding Behaviors toward Developing Selective Inhibitors of the Peripheral Endocannabinoid System for the Treatment of Obesity and Metabolic Syndrome

**DOI:** 10.3390/toxins11050275

**Published:** 2019-05-15

**Authors:** Shira Hirsch, Joseph Tam

**Affiliations:** Obesity and Metabolism Laboratory, Institute for Drug Research, School of Pharmacy, Faculty of Medicine, The Hebrew University of Jerusalem, Jerusalem 9112001, Israel; shirah@ekmd.huji.ac.il

**Keywords:** *Cannabis*, marijuana, CB1 receptor, central CB1 receptor blockade, peripheral CB1 receptor blockade

## Abstract

In this review, we discuss the role of the endocannabinoid (eCB) system in regulating energy and metabolic homeostasis. Endocannabinoids, via activating the cannabinoid type-1 receptor (CB_1_R), are commonly known as mediators of the thrifty phenotype hypothesis due to their activity in the central nervous system, which in turn regulates food intake and underlies the development of metabolic syndrome. Indeed, these findings led to the clinical testing of globally acting CB_1_R blockers for obesity and various metabolic complications. However, their therapeutic potential was halted due to centrally mediated adverse effects. Recent observations that highlighted the key role of the peripheral eCB system in metabolic regulation led to the preclinical development of various novel compounds that block CB_1_R only in peripheral organs with very limited brain penetration and without causing behavioral side effects. These unique molecules, which effectively ameliorate obesity, type II diabetes, fatty liver, insulin resistance, and chronic kidney disease in several animal models, are likely to be further developed in the clinic and may revive the therapeutic potential of blocking CB_1_R once again.

## 1. Overview of Plant Cannabinoids and Endocannabinoids

Throughout human history, plants have been used as a predominant source of medications. The genus *Cannabis* includes up to three strains, *Cannabis sativa*, *Cannabis indica*, and *Cannabis ruderalis,* each with a very long history of domestication [1]. These strains can be separated by morphology, by phytochemistry, and by differences in their original geographic area. Hybrid variations of these strains have been developed to strengthen some specific characteristics in order to make cannabis an effective drug [2]. Regarding its unique chemistry, *Cannabis sativa* (marijuana) is considered one of the most resourceful plants, research investigations of which during the past half-century have led to the discovery of an important homeostatic system, the endocannabinoid (eCB) system, which plays a key role in human physiology (reviewed in [3]). Currently, 545 natural compounds have been identified from this plant [4]. Of these, 144 have been isolated and identified as cannabinoids (phytocannabinoids) [5]. The first attempt to successfully identify a cannabinoid was made in 1899 by Wood and colleagues [6], who isolated cannabinol (CBN). However, it took almost forty years and several groups’ efforts to identify the correct structure of CBN (reviewed in [7]). Interestingly, the most advanced characterization of different phytocannabinoids was done during the 1960s by Mechoulam’s group, who isolated and reported the correct structure and stereochemistry of cannabidiol (CBD) [8], Δ^9^-tetrahydrocannabinol (Δ^9^-THC, the main psychoactive component of marijuana) [9,10], Δ^8^-tetrahydrocannabinol (Δ^8^-THC) [11], cannabigerol (CBG) [12], cannabichromene (CBC) [13], and cannabicyclol (CBL) [14].

Since then, three decades had passed until the binding sites of Δ^9^-THC in the brain and peripheral organs were identified, which were then termed as the cannabinoid-1 and -2 receptors (CB_1_R and CB_2_R, respectively) [15,16,17]. As of recently, their structures have been cloned and reported by several groups [18,19,20,21]. These characterizations will significantly aid in developing more specific synthetic cannabinoids in the future. Signaling by both receptors is mainly mediated via G_i_/G_o_ proteins, despite the fact that they can also recruit G_s_ and G_q/11_ proteins and facilitate G protein-independent molecular pathways [22]. CB_1_R, primarily localized in the cell membrane, is the most widely expressed G-protein coupled receptor (GPCR) in the human brain [23], but it is also abundantly expressed in peripheral organs [24]. CB_2_R, on the other hand, is predominantly localized in immune cells and is moderately expressed in many peripheral tissues, with conflicting evidence regarding its expression in the central nervous system (CNS) [25]. Of the 144 phytocannabinoids present in *Cannabis*, only Δ^9^-THC and its less abundant propyl analogue, Δ^9^-tetrahydrocannabivarin (THCV), have been shown to bind to CB_1_R and CB_2_R with high affinity (with agonistic and antagonistic activity for THC and THCV, respectively). Regarding other cannabinoids, studies have shown their ability to bind to several different receptors, ranging from other GPCRs (GPR18, GPR55, and GPR119) to ion channel (thermosensitive transient receptor potential (TRP) channels) and nuclear receptors (peroxisome proliferator-activated receptors, PPARs) (reviewed in [26]); however, their physiological functions are still largely unknown.

The successful cloning and identification of CB_1_R and CB_2_R in mammalian cells prompted the discovery of their first endogenous ligand, arachidonoyl ethanolamide (AEA, or anandamide) [27], which was then followed by identifying 2-arachidonoyl glycerol (2-AG) [28,29]. Whereas AEA is a high-affinity, partial agonist of CB_1_R, and barely active at CB_2_R, 2-AG is known to activate both receptors with moderate-to-low affinity [30,31]. Both eCBs are synthesized, transported, and inactivated in their respective target tissues differently. Whereas AEA is catalyzed from *N*-acyl-phosphatidylethanolamine (NAPE) by NAPE-specific phospholipase D (NAPE-PLD) or via other means [3], 2-AG is mainly generated from diacylglycerol (DAG) by either DAG lipase (DAGL) α or β [32]. Their degradation depends on the specific cellular uptake and enzymatic catabolism. AEA is degraded primarily by membrane-associated fatty-acid amide hydrolase (FAAH) into free arachidonic acid and ethanolamine [33], whereas 2-AG is predominantly hydrolyzed by monoglyceride lipase (MAGL) into arachidonic acid and glycerol [34].

The eCB system, acting both centrally and peripherally, is an important physiological system, comprising the cannabinoid receptors and their natural endogenous ligands as well as the enzymes/proteins involved in their biosynthesis, transport, and degradation. It is involved in many physiological and pathological conditions and functions as a regulatory homeostatic system in various tissues, such as the brain, skin, liver, cardiovascular system, bone, kidney, pancreas, adipose and muscle tissues, the digestive track, and many more (reviewed in [3]). Since it is ubiquitously present in humans and animals, it has been suggested that its homeostatic roles are “relax, eat, drink, rest, sleep, save, store, forget, and protect” [35]. Therefore, changes in eCB ‘tone’, represented by the expression of the cannabinoid receptors, their functional activity (upregulated or downregulated), and the relative amount of eCBs, may render the subject susceptible to different diseases. For instance, enhanced eCB ‘tone’ has been linked to the development of many metabolic diseases (e.g., obesity, type II diabetes, fatty liver disease, and chronic kidney diseases) [24], whereas reduced eCB ‘tone’, also termed ‘clinical eCB deficiency syndrome’, is associated with migraine, fibromyalgia, irritable bowel syndrome, schizophrenia, multiple sclerosis, Huntington’s, Parkinson’s, anorexia, chronic motion sickness, and autism [36,37,38]. Therefore, utilizing different approaches to achieve modulatory effects on the eCB system and ‘normalize’ its action under these conditions (by using various phytocannabinoids, synthetic cannabinoids, and novel drugs that may affect eCB ligand synthesis or degradation) is advised.

## 2. Is Marijuana a Toxic Drug?

Cultivated for millennia, marijuana still has a remarkable ability to alleviate different physical pathologies. To date, the U.S. Federal Food, Drug, and Cosmetic Act defines marijuana as a drug, taken by either smoking or consuming it orally for therapeutic purposes. As a drug, it may also cause harm and has toxic effects. Thus far, only limited reports related to the side effects of marijuana used for medical purposes have been reported. This contradicts the existing information regarding its recreational use, as well as in comparison to other drugs (e.g., morphine and cocaine). Whereas the latter drugs may cause death when consumed inappropriately, mainly due to respiratory arrest [39,40] and/or increasing the blood pressure and heart rate [41,42], no such evidence has been reported with the use of marijuana. Nevertheless, a few generalized findings are related to the acute and chronic side effects of cannabis use. Among them, cannabinoids have been shown to affect: (i) the cardiovascular system (acute use is associated with tachycardia and increased blood pressure vs. chronic exposure that results in the opposite effects); (ii) the respiratory system, in which inflammation of the lungs and large airways is increased; bronchitis and emphysema have been documented with the chronic use of cannabis; (iii) cognition, by reducing attention, sensory perception, task acquisition, and working memory; and (iv) mental illness and psychiatric conditions, including depression, anxiety, psychosis, bipolar disorder, and schizophrenia (summarized in [43,44,45,46]).

## 3. “To Eat or Not to Eat”: The Role of Cannabinoids in Feeding Behaviors

Although not toxic, a common ‘side effect’ of cannabis use is an increase in appetite. This well-known property, coupled with the existence of CB_1_R within appetite-related brain areas [47], suggests that the eCB system plays a key role in regulating feeding and body weight. For centuries, marijuana has been recognized as a food intake stimulant. Although the first evidence of cannabis use for treating appetite loss was reported in 300 A.D. in India, a few studies conducted in humans during the 20^th^ century firmly supported the ability of cannabis consumption to induce hyperphagia and snacking (collectively referred to as ‘the munchies’; summarized in [48]). Indeed, marijuana use in healthy normal volunteers has been shown to increase daily caloric intake, which is mainly because of enhanced food intake between meals rather than an increase in meal size [49]. Orally administered Δ^9^-THC or cannabis smoking enhances the consumption of highly palatable and sweet snack foods and increases the qualitative ratings of hunger [49,50,51], findings that support the role of the eCB/CB_1_R system in regulating feeding behaviors via the reward system [52]. These cumulative data actually support the clinical evaluation and testing of cannabinoid therapeutics to stimulate appetite in cancer patients undergoing chemotherapy [53], individuals with HIV/AIDS [54,55,56,57], and anorexia nervosa [58] as well as in anorexic Alzheimer’s disease patients [59,60].

Accumulating basic evidence also supports the orexogenic effects of cannabinoids, demonstrating increased food intake by administering Δ^9^-THC in various animal models [61,62,63,64,65]. However, our current understanding of cannabinoid action on food intake was revolutionized after CB_1_R was identified in various brain regions, including the hypothalamus, which plays a key role in homeostatic regulation. Indeed, direct activation of CB_1_R by AEA has been shown to stimulate food intake [66,67,68,69]. Other CB_1_R agonists were also reported to increase sucrose consumption [64,70] and hyperphagia [71]. An interesting observation reporting hypophagia induced by high doses of Δ^9^-THC was reported in 1975 [72]. In fact, Chopra and Chopra reported in 1939 that while a weak cannabis preparation stimulates appetite, more potent cannabis preparations usually have an opposite effect [73]. Similarly, Bouquet also noted that progressive anorexia develops with chronic use of cannabis [74]. Further studies conducted in animals confirmed that cannabinoid administration in high doses induces hypophagia (summarized in [75]). Data on cannabis use, caloric intake, and body mass index (BMI) establish conclusive evidence that chronic cannabis use is associated with reduced BMI and obesity rates (summarized in [76]). Interestingly, despite having a lower BMI, most cannabis users appear to have increased caloric intake. This paradox can be causatively explained by the fact that heavy cannabis use results in downregulation of CB_1_R [77,78,79], which in turn may lead to weight loss [76]. In keeping with this explanation, in recent years, many studies have examined how antagonizing CB_1_R affects feeding behavior and subsequently induces weight loss in obese individuals.

## 4. Targeting CB_1_R for Treatment of Obesity: Block Centrally or Inhibit Peripherally

Empirical studies in various animal models indicate that pharmacological blockade of CB_1_R with the first-in-class synthetic CB_1_R inverse agonist rimonabant (SR141716A) does indeed reduce weight gain and food intake in a dose-dependent manner under both fasted and non-fasted conditions [61,80,81,82,83] as well as inhibit the motivation for palatable food [84,85]. These data, together with the fact that animals genetically lacking CB_1_R are hypophagic and lean [86], led to the idea that CB_1_R blockade could be considered as a therapeutic tool against obesity and metabolic syndrome. Indeed, rimonabant was proven effective not only in decreasing food intake and body weight, but also in ameliorating obesity-induced insulin and leptin resistance, improving glucose homeostasis and dyslipidemias, as well as decreasing hepatic steatosis in obese/overweight individuals with metabolic syndrome [87,88,89,90,91,92,93]. These clinical studies led to the approval of rimonabant by the European Medicines Agency (EMA) in 2006 as an antiobesity drug under the name of Acomplia^®^ (Sanofi-Aventis). However, growing evidence of anxiety, depression, and suicidal ideation, which was reported in a small but significant portion of individuals treated with rimonabant [94], led to its eventual withdrawal from the market in 2009. This decision affected all the big pharmaceutical companies that were developing their own CB_1_R blockers, and questions were raised regarding the therapeutic relevance of this class of molecules in modulating the eCB system for treatment of metabolic syndrome [95].

Despite having only a transient inhibitory effect on feeding, rimonabant was very efficacious in reducing body weight and adiposity, suggesting that CB_1_R blockade not only affects CNS-mediated energy homeostasis, but also regulates energy balance via peripheral mechanisms [96]. As mentioned before, CB_1_Rs are present not only in the CNS, but also in many peripheral organs. Their expression levels in adipose tissue, liver, skeletal muscle, kidney, and pancreas are elevated under obese/diabetic conditions [97,98,99,100,101,102,103]. A parallel elevation in tissue and circulating eCB levels in obesity has also been vastly documented [100,102,104,105,106,107,108,109,110,111]. By utilizing several genetic models with a specific deletion of CB_1_R in liver, adipose tissue, kidney, pancreas, and skeletal muscle, studies have shown that CB_1_R modulates peripheral metabolic function. Interestingly, deletion of hepatic CB_1_R was sufficient to protect obese mice from hepatic steatosis and dyslipidemia, as well as insulin and leptin resistance [100]. A specific deletion of CB_1_R in adipocytes resulted in complete protection from diet-induced obesity in mice [112]. Beta cell-specific CB_1_R-knockout mice are protected from high-fat/high-sugar diet-induced pancreatic dysfunction and inflammation [113], and its specific ablation from skeletal muscle protects mice from diet- and age-induced insulin resistance [114]. Recently, we have shown that diabesity-induced renal abnormalities are mediated via CB_1_R specifically located on the renal proximal tubule cells (RPTCs) [102,115,116]. Whereas obese or diabetic mice lacking CB_1_R in the RPTCs gain weight and show metabolic impairment similar to their wild-type control animals, they remain completely protected from diabesity-induced renal dysfunction, inflammation, fibrosis, lipotoxicity, and mitochondrial function [102,115,116]. Taken together, the apparent increase in peripheral eCB ‘tone’ in obesity and the key role CB_1_R plays in cellular/metabolic regulation in many peripheral organs suggest that targeting CB_1_R in peripheral organs by limiting brain access of CB_1_R blockers may improve their therapeutic efficacy via reducing their potential to cause CNS-mediated adverse effects. This idea was tested experimentally in numerous studies describing the contribution of the peripheral eCB/CB_1_R system to the development of obesity and its metabolic comorbidities, as well as the therapeutic potential of peripherally restricted CB_1_R antagonists to treat obesity and its sequelae.

## 5. Current View Regarding Novel Peripherally Restricted CB_1_R Blockers

Identifying novel and robust peripherally restricted CB_1_R antagonists devoid of brain penetration and CNS activity can be achieved by using two main paradigms: First, chemical modification of brain-penetrating CB_1_R blockers, such as rimonabant or other rimonabant-like compounds (such as taranabant, otenabant, ibipinabant, etc.); second, usage of computational or in vitro chemical tools to design and synthesize compounds that do not penetrate the blood–brain barrier (BBB), based on studies that characterize those properties responsible for brain penetration [117]. In both models, one should take into consideration the physicochemical properties (e.g., lipophilicity, hydrogen bonding capacity, molecular weight, and polar surface area) required for brain restriction, as well as the usage of efflux transporters, which may also depend on the compound’s structure. The preferred conditions for peripherally restricting CB_1_R blockers are well-described elsewhere [118]. In brief, such a compound needs to be less hydrophobic and more polar in nature to make it impenetrable into the CNS, two properties that mainly govern passive diffusion of a molecule through the BBB [119,120]. To date, various novel molecules with peripheral selectivity toward CB_1_R and limited BBB penetration have been designed and patented by different groups (summarized in [121]; Table 1). Only those that have been characterized and tested experimentally against obesity are highlighted in the following paragraphs.

AM6545 was the first to undergo a detailed pharmacological, metabolic, and behavioral assessment in murine models of obesity. This molecule ameliorates hepatic steatosis, increases insulin sensitivity, and improves dyslipidemia in diet- and genetically induced obese mice [122]. In addition, AM6545 has been shown to reduce food intake, the meal size, the rate of feeding, and body weight in obese animals [123,124,125]; attenuate obesity-induced dyslipidemia via activating brown adipose tissue [126]; and reverse monosodium glutamate-induced hypometabolic and hypothalamic obesity in mice [127]. Soon after, another well-characterized novel peripherally restricted CB_1_R antagonist, JD5037, was developed and preclinically tested against obesity. JD5037 was found to be equally efficacious in reducing body weight and food intake, improving glycemic control, and attenuating hepatic steatosis with its brain-penetrating parent compound SLV319 (Ibipinabant^®^) [128]. Its hypophagic role is most likely mediated via increasing hypothalamic leptin sensitivity, although it is inactive on brain CB_1_Rs [128,129]. More recently, JD5037 was found to reduce hyperphagia and weight gain in *Magel2* null mice, a well-established model of Prader–Willi syndrome [109], as well as to reverse fatty acid flux-, CB_1_R-, and type I diabetes-induced renal impairment [102,116].

A few other novel molecules that mostly target CB_1_R in the periphery have also been synthesized and characterized, although not to the same extent as AM6545 and JD5037. Among these, TM38837, also recently termed BPR0912, has a negligible impact on brain CB_1_R when tested in mice, primates, and healthy individuals [130,131,132], and has been shown to decrease body weight in rodents [133,134] and improve the cardiometabolic complications associated with obesity via increasing thermogenesis in white and brown adipose tissues [135]. NESS06SM, a peripherally selective CB_1_R neutral antagonist whose structure is related to rimonabant, was found to be efficacious in ameliorating diet- and olanzapine-induced obesity and its metabolic abnormalities [136,137]. LH-21, initially considered as a neutral peripherally restricted CB_1_R blocker able to reduce food intake and body weight in rats [138,139,140], was recently found to penetrate the BBB and reduce food intake in CB_1_R null mice [141]. URB447 lowers food intake and body weight in mice [142], probably via reducing fat ingestion through the gut [143,144] in a CB_1_R-dependent manner. With an IC_50_ value of 159 nM, Compound 1, described by Son and colleagues in 2010, was found to be less brain-penetrating and efficacious than rimonabant in ameliorating food intake and obesity in mice [145]. With a considerably lower exposure in the brain, Compound D4, developed by 7TM Pharma, induced pronounced weight reduction in a dose-dependent manner in obese mice in comparison with rimonabant [146]. Although designed to be a P-glycoprotein (P-gp) substrate in order to decrease its brain penetration, Compound 6a, developed by Janssen Research & Development, accumulated in the brain following chronic administration, suggesting that its in vivo metabolic efficacy cannot exclude blocking central CB_1_Rs [147]. Compound 2p, which originated from the brain-penetrant CB_1_R inverse agonist program of the same group, reduced glucose levels without centrally mediated behavioral effects and a reduction in food intake or body weight [148]. Lastly, TXX-522, a newly synthesized compound that exhibited minimal brain penetration while retaining high affinity and selectivity toward CB_1_R, improved dyslipidemia, glucose homeostasis, and fat mass in obese mice without affecting their food intake [149]. Overall, CB_1_Rs located in the periphery can be potentially considered as clinically relevant targets for therapeutics against obesity and its comorbidities, thus warranting further preclinical development and clinical testing of the peripherally restricted CB_1_R blockers. Of note, in December 2017, the U.S. Food and Drug Administration (FDA) cleared the Investigational New Drug (IND) Application for JD5037 to begin Phase 1 clinical trials. However, it remains to be seen if this novel compound will clear the way for other molecules that target peripheral CB_1_Rs to be fully translated into use for humans and rekindle the spark for discovering new blockbuster therapies against metabolic syndrome.

## 6. Concluding Remarks

The appetite-stimulating ‘side effect’ of marijuana has been recognized for centuries. Mounting evidence supports the key role that CB_1_Rs play in orexigenic signaling via central modulation of energy balance and feeding behavior. However, the influence of the eCB/CB_1_R system on energy utilization and homeostasis cannot be solely explained by central mechanisms. Indeed, data show that this system also acts peripherally to modulate adipose tissue metabolism, kidney function, hepatic lipogenesis, muscle activity, and pancreatic homeostasis. Being tonically overactivated during obesity, the eCB/CB_1_R system contributes to impairment in hormonal/metabolic function, propelling CB_1_R forward as a potential therapeutic target for obesity. Whereas the globally acting CB_1_R blocker rimonabant once held tremendous promise in ameliorating the metabolic abnormalities of obesity, its CNS-mediated adverse effects limited its clinical use. Targeting the eCB system using novel compounds that block CB_1_Rs in periphery with negligible brain penetration still holds promise for future therapy for obesity and its sequelae.

## Figures and Tables

**Table 1 toxins-11-00275-t001:** List of peripherally restricted cannabinoid type-1 receptor (CB_1_R) antagonists.

Compound	CB_1_R Ki/EC_50_/IC_50_	CB_2_R Ki/EC_50_/IC_50_	Nature of Compound	cLogP/LogP	TPSA/PSA (Å^2^)	HBD	Animal Model	Efficacy	Brain/Plasma Ratio	Structure	Ref.
**LH-21**	EC_50_ = 76.9 nM	EC_50_ = 6.56 µM	Neutral antagonist	N/A	N/A	N/A	Obese and lean Zucker rats	Reduces food intake, no change in lipid level and plasma glucose	N/A	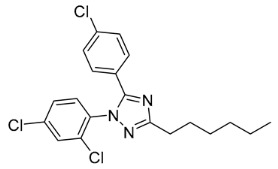	[138,139,140,141]
**URB447**	IC_50_ = 313 nM	IC_50_ = 41 nM	Neutral antagonist (CB_1_R)/ agonist (CB_2_R)	LogP = 6.39	PSA = 48.02	N/A	*ob/ob* mice	Reduces food intake and body weight gain	N/A	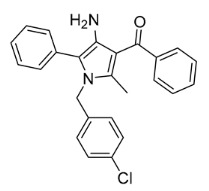	[142,143,144]
**AM6545**	Ki = 3.3 nM	CB_1_R/CB_2_R > 100	Neutral antagonist	LogP = 3.3	PSA = 116	1	DIO C57BL/6 mice	Reduces body weight, hepatic triglyceride content, and hepatocellular damage; increases fat oxidation	0.03	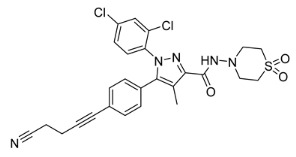	[122,123,124,125,126,127]
**Compound 1**	IC_50_ = 159 nM	>10 µM	Antagonist	N/A	N/A	N/A	DIO C57BL/6 mice	Reduces body weight and suppresses DIO-induced elevation in hepatic SREBP-1 expression	CLapp., uptake = 0.00228	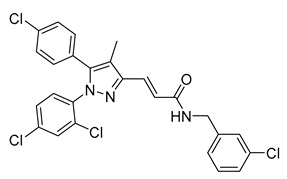	[145]
**Compound D4**	IC_50_ = 2.6 nM	CB_1_R/CB_2_R > 1000 nM	Antagonist	N/A	N/A	N/A	DIO C57BL/6 mice	Reduces body weight	0.098	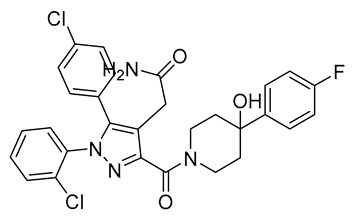	[146]
**TM38837 (BPR0912)**	IC_50_ = 8.5 nM EC_50_ = 18.5 nM	IC_50_ = 605 nM	Antagonist	LogP = 8.91	TPSA = 78	1	DIO C57BL/6 mice	Decreases body weight and increases thermogenesis	0.03	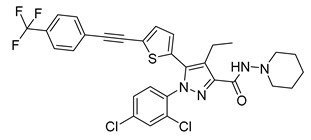	[130,131,132,133,134,135]
**JD5037**	Ki = 0.35 nM	CB_1_R/CB_2_R > 700 nM	Inverse agonist	cLogP = 6	PSA = 117	3	DIO C57BL/6 mice	Reduces food intake, body weight, and improves hormonal/ metabolic abnormalities	0.02	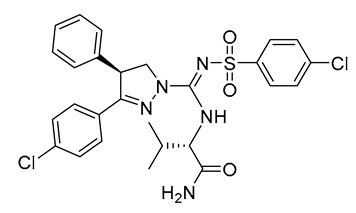	[102,109,116,128,129]
**Compound 14h**	Ki = 5.1 nM	Ki > 10,000 nM	Antagonist	LogP = 3.7	N/A	N/A	DIO Sprague−Dawley rats	No metabolic effect	0.13	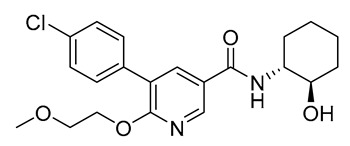	[150]
**NESS06SM**	Ki = 10.25 nM	Ki > 5000 nM	Neutral antagonist	cLogP = 4.62	TPSA = 59.39	N/A	DIO C57BL/6 mice	Reduces body weight and visceral fat mass, improves blood glucose and dyslipidemia	logBB = −0.038 (low)	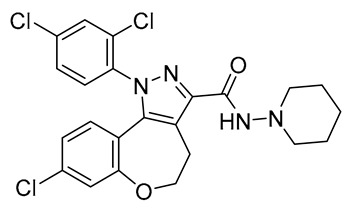	[136,137]
**Compound 2p**	EC_50_ = 0.035 µM	EC_50_ = 2.0 µM	Inverse agonist	cLogP = 7.27	TPSA = 59.8	N/A	DIO C57BL/6 mice	Lowers plasma glucose levels	0.05	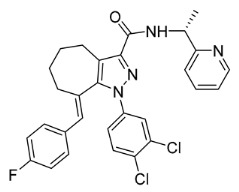	[148]
**Compound 8c**	Ki = 8.82 nM	Ki = 1545 nM	Inverse agonist	N/A	TPSA = 76	N/A	N/A	N/A	0.15	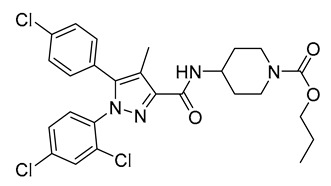	[151]
**TXX522**	IC_50_ = 10.33 nmol/L	IC_50_ > 10 µmol/L	Neutral antagonist	LogP = 7.95	TPSA = 56.73	1	DIO C57BL/6 mice	Reduces body weight and fat mass, decreases metabolic complications	0.02 (Kp)	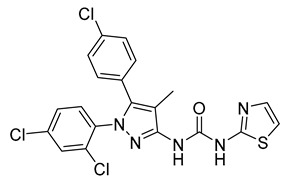	[149]
**Compound 6a**	EC_50_ = 0.0082 µM	EC_50_ > 10 µM	Inverse agonist	cLogP = 6.15	TPSA = 86.9	2	DIO C57BL/6 mice	Reduces body weight, food intake, insulin level, liver fat, and cholesterol	0.027	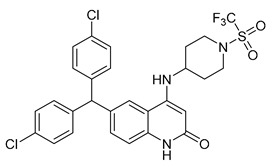	[147]
**Compound 65**	Ki = 4.0 nM	Ki > 10,000 nM	Inverse agonist	N/A	N/A	N/A	N/A	N/A	0.18	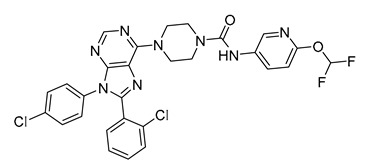	[152]
**AJ5018**	IC_50_ = 90.4 nM	N/A	Antagonist	N/A	N/A	N/A	DIO C57BL/6 and *db/db* mice	Reduces hyperglycemia, dyslipidemia, hepatic steatosis, energy expenditure, and insulin resistance	0.1	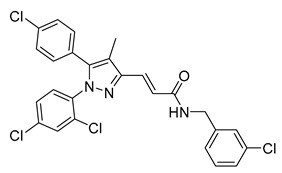	[153]
**AJ5012**	N/A	N/A	Antagonist	AlogP = 5.328	PSA = 84.836	N/A	DIO C57BL/6 and *db/db* mice	Reduces weight, increases energy expenditure; improves metabolic abnormalities, glycemic control, and insulin sensitivity	0.2	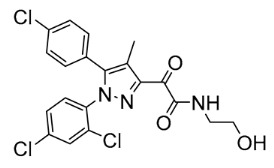	[154]
**Compound 17a**	Ki = 47.1 nM	Ki = 20,000 nM	Antagonist	N/A	TPSA = 79		Sprague Dawley rats	N/A	0.0320	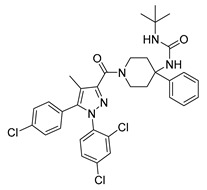	[155]
**Compound 18a**	Ki = 2.9 nM	Ki = 2510 nM	Antagonist	N/A	TPSA = 76		Sprague Dawley rats	N/A	0.0214	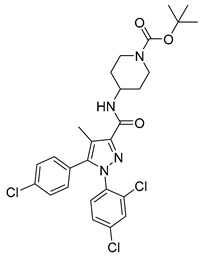	[155]
**Compound 18f**	Ki = 14.7 nM	Ki = 3349 nM	Antagonist	N/A	TPSA = 79		Sprague Dawley rats	N/A	0.379	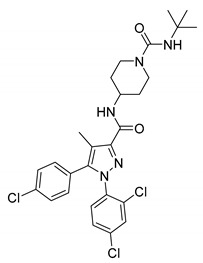	[155]
**ENV-2**	N/A	N/A	Antagonist	N/A	N/A	N/A	Wistar rats	Reduces glycemia and dyslipidemia	N/A	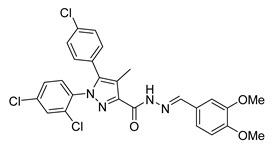	[156]
**MJ08**	Ki = 25.4 nM IC_50_ = 99.9 nmol/L	N/A	Inverse agonist	N/A	N/A	N/A	Wistar rats, DIO C57BL/6 mice	Stimulates hepatic glucose production	N/A	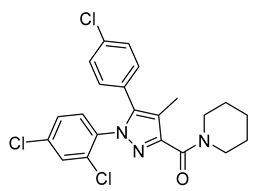	[157,158]
**PISMR**	Ki = 57 nM	N/A	Antagonist	N/A	N/A	N/A	DIO C57Bl/6 mice	Reduces weight, food intake, and adiposity as well as improving glycemic control and lipid homeostasis	0.24	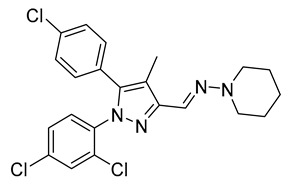	[159,160]

Half maximal effective concentration (EC_50_); Half maximal inhibitory concentration (IC_50_); Cannabinoid type-2 receptor (CB_2_R); Calculated Log P (cLogP); Topological polar surface area/Polar surface area (TPSA/PSA); Hydrogen bond donor (HBD); Not available (N/A); Diet-induced obese (DIO); Apparent brain uptake clearance (CLapp); Ratio of the steady-state concentrations of the drug molecule in the brain and in the blood, expressed as log (C_brain_/C_blood_; logBB); Brain to plasma distribution ratio (Kp); Atom-based Log P (ALogP).

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
