# Peer review of "Cannabis: From a Plant That Modulates Feeding Behaviors toward Developing Selective Inhibitors of the Peripheral Endocannabinoid System for the Treatment of Obesity and Metabolic Syndrome"

_toxins, 2019, doi:10.3390/toxins11050275_

Round 1
Reviewer 1 Report
The authors present an overall review of past and current research conducted on the utility of endocannabinoid modulation as a therapeutical antiobesity approach. They give a brief overview of the discovery and characterization of the endocannabinoid system. The physiological significance of the endocannabinoid system and the clinical implications of changes in the tone of the system are given. The dual role of cannabinoids in feeding behaviors are discussed, with supporting evidence. The authors then go on to discuss the role of pharmacological blockades of CB1R in the treatment of obesity, specifically peripherally restricted blockers. An overview of current research is provided to justify further research on a promising target for the treatment of obesity.
Comments:
The authors provide a clearly laid out review of the current literature. The review is presented in a manner which is clear for a non-expert reader, and the story behind current knowledge is presented in a logical and interesting fashion. The authors include many recent publications to support their hypothesis. This paper provides a valuable and comprehensive overview of current knowledge in a digestible form likely to be of interest to many readers.
The authors do mention Cannabis sativa, but give no mention of other strains. The paper could benefit from acknowledging that other strains do exist and are in use.
Author Response
Thank you for finding our manuscript valuable and comprehensive.
We added a short description of the three major strains of Cannabis and cited two recent papers that further discuss this matter (references #1 & 2). Please see the revised text on page 1, lines 31-36.
Reviewer 2 Report
General comment
This short review provides an interesting, well documented perspective upon the use of the inhibitors of cannabinoid type-1 receptors (CB1Rs) as a possible strategy to counteract the pathogenesis of obesity and metabolic syndrome. The authors discuss synthetically the regulatory role of CB1Rs of central nervous system (CNS) in feeding behavior (mainly reflecting the properties of cannabis, at low-moderate doses, as a food intake stimulant) and of peripheral CB1Rs in metabolism and energy balance in several organs and tissues, such as adipose tissue, muscle, liver, kidneys and pancreas. The mechanisms by which C1BRs regulate energy balance in peripheral organs of individuals affected by the metabolic syndrome include the amelioration of obesity-induced insulin and leptin resistance, improvements in glucose homeostasis and dyslipidemias, and the reduction of hepatic steatosis. Emphasis is placed in the manuscript onto novel drugs that act mostly on CB1R in peripheral organs and have been proven to be able to ameliorate obesity, insulin resistance, type II diabetes, fatty liver and chronic kidney disease, by counteracting the enhanced ‘endocannabinoid tone’ which is found in these diseases. Limited penetration in the brain by these ‘peripheral’ C1BR blockers should ensure avoidance of adverse effects on CNS, such as anxiety, depression and suicidal behavior, which were displayed instead by early pan-CB1R inhibitors. The authors briefly discuss some strategies currently employed to develop peripherally restricted CB1R blockers and provide an informative perspective of the results obtained with the latter, mostly in experimental animals. In conclusion, the review warrants continued pre-clinical and clinical testing of the therapeutic potential of peripheral CB1R blockers in obesity and metabolic syndrome.
Overall, the review is timely, informative, logical, clear and well-written. Table 1 is very informative. I only have one major and a few minor specific remarks, which follow.
Major specific point
Title: Even though one may understand that the title of a manuscript has to be somewhat ‘spicy’, in order to attract the interest of potential readers, I do think that the present title of this manuscript is somewhat misleading. In its present form, the title seem to emphasize the antiobesity properties of cannabis (which are only manifested in association with the chronic consumption of high doses of the drug, together with other undesirable chronic side-effects), whereas further reading makes it clear that the interest for the fight against metabolic diseases and obesity is in phytocannabinoids or synthetic cannabinoids which display selective inhibitory action on peripheral CB1Rs and, thus, may counteract the enhanced ‘eCB’ tone which accompanies these conditions. In my opinion, a better title should be: “Therapeutic potential of selective inhibitors of peripheral endocannabinoid receptors in obesity and metabolic syndrome”.
Minor comments
Lines 15, 76, 211: please amend ‘diabetes’ to ‘type II diabetes’
Lines 26-27: please move here one or a few of the reference(s) to general review(s) on the endocannabinoid system that are cited further on in the text
Line 98: please insert a comma after the word ‘cognition’
Lines 180 and 256: please amend ‘sequela’ to ‘sequelae’
Line 226: please amend ‘induces’ to ‘induced’
Line 229: please amend ‘accumulates’ to ‘accumulated’
Line 232: please amend ‘reduces’ to ‘reduced’
Author Response
We would like to thank the reviewer for finding our review suitable for publication following addressing his/her comments.
Thank you for this major comment. We do understand how our title may somehow mislead the readers and we do appreciate the suggested title. Since our review was invited to participate in a special issue of Toxins "From Toxins to Drugs", we wanted to highlight the pharmacological rationale of developing drugs that block the CB1R based on the fact that low doses of cannabis increase feeding and high doses suppress it. Nevertheless, we accepted the comment and amended the title accordingly. The new title is: Cannabis: From a Plant that Modulates Feeding Behaviors toward Developing Selective Inhibitors of the Peripheral Endocannabinoid System for the Treatment of Obesity and Metabolic Syndrome
All minor comments were addressed. Thank you.
Reviewer 3 Report
The manuscript reviews the current state of research into one of the side effects of cannabinoid use - Hyperphagia and its potential for therapeutic application. It provides a concise review of the key considerations into development of cannabinergic drugs for the treatment of obesity. The authors briefly review the cannabinoid system, the failure of non-targeted CB1 blockers and the recent trend in the field to focus on targeting peripheral cannabinoid mechanisms. The authors provide a comprehensive list of the current crop of peripherally-acting cannabinoid antagonists reported in the literature. This could prove to be a valuable resource for understanding the properties of current list of compounds and also for further development of next generation of drug candidates.
The manuscript is generally well written with appropriate references. I would recommend its publication following the correction of a couple of spelling errors mentioned below -
1) Line 84 - Marijuana spelling
2) Line 109 - Snacking spelling
Author Response
We would like to thank the reviewer for finding our review appropriate for publication. The spelling errors have been corrected. Thank you.
Reviewer 4 Report
The present manuscript is one of the many attempts these days to summarize the current status of cannabinoid system in obesity. However, by focusing on peripherally restricted cannabinoid ligands, this paper brings an interesting perspective and deserves to be published.
Still, I have two major demands:
1) Table S1 must be included in the main text as it includes the most important part of the paper.
2) Please change the title, as 'munchies' is not funny and does not describe at all the effects of cannabinoids on body homeostasis.
Author Response
Thank you for finding our manuscript suitable for publication.
Table S1 is now included in the main text and not as a Supplementary Table.
The title has been amended also according to the request of Reviewer #2. The new title is: Cannabis: From a Plant that Modulates Feeding Behaviors toward Developing Selective Inhibitors of the Peripheral Endocannabinoid System for the Treatment of Obesity and Metabolic Syndrome